# Preparation of an Asymmetric Membrane from Sugarcane Bagasse Using DMSO as Green Solvent

**Dang Thi To Nu [1], Nguyen Phi Hung [1], Cao Van Hoang [1] and Bart Van der Bruggen [2,3,*]**

1   Department of Chemistry, Quy Nhon University, Quy Nhon 590000, Vietnam
2   Laboratory for Process Engineering for Sustainable Systems, Department of Chemical Engineering, KU Leuven, Celestijnenlaan 200F, 3001 Leuven, Belgium
3   Faculty of Engineering and the Built Environment, Tshwane University of Technology, Private Bag X680, Pretoria 0001, South Africa
*   Correspondence: bart.vanderbruggen@kuleuven.be; Tel.: +32-16-322340

**Abstract:** Asymmetric cellulose acetate membranes have been successfully fabricated by phase inversion, using sugarcane bagasse (SB) as the starting material. SB is a raw material with high potential to produce cellulose derivatives due to its structure and morphology. Cellulose was extracted from SB by pretreatment with solutions of 5 wt% NaOH, 0.5 wt% EDTA; then bleached with 2 wt% $H_2O_2$. Cellulose acetate (CA) was prepared by the reaction between extracted cellulose with acetic anhydride, and $H_2SO_4$ as a catalyst. The obtained CA exhibited a high degree of substitution (2.81), determined with $^1$H-NMR spectroscopy and titration. The functional groups and thermal analysis of the extracted cellulose and the synthesized CA have been investigated by Fourier transform infrared spectroscopy (FT-IR), differential scanning calorimetry (DSC), and thermogravimetric analysis (TGA). The change in the crystallinity of the extracted cellulose and CA was evaluated by X-ray diffraction (XRD) spectroscopy. Asymmetric membranes were fabricated using dimethyl sulfoxide (DMSO) as the solvent, with a casting thickness of 250 μm. The obtained membranes were studied by scanning electron microscopy (SEM), DSC and atomic force microscopy (AFM). The hydrophilicity of the membranes was evaluated, as demonstrated by the measurement of water contact angle (WCA) and water content. Furthermore, the antifouling properties of membranes were also investigated.

**Keywords:** sugarcane bagasse; cellulose acetate; DMSO; phase inversion; asymmetric membrane

## 1. Introduction

Today, sustainable development is a particular concern, in which the search for renewable sources aims at developing biodegradable materials with interesting mechanical properties. Lignocellulosic biomass is the most abundant natural cellulose source on earth. Within this source, sugarcane bagasse (SB) is widely available in tropical and subtropical areas [1]. Sugarcane bagasse, which is the solid and fibrous fraction remaining from milling sugarcane, contains mainly cellulose (30–50%), hemicellulose and lignin [2]. The conversion of cellulose into economic highly-valued derivative compounds has drawn widespread attention in the field of green and sustainable chemistry, and stimulated the development of environmentally friendly technologies [3,4]. High-purity cellulose is suitable for the synthesis of cellulose derivatives. Among the cellulose esters, cellulose acetate (CA) is the most important one because of its biodegradability, low cost, non-toxicity, and being a renewable source [5]. Due to these advantages, CA is extensively used in the preparation of membranes, fibers, films, and biomedical utilities [2,6,7]. CA is usually synthesized from acetic acid and acetic anhydride in the presence of sulfuric acid as a catalyst and with a degree of substitution depending on experimental conditions [8]. The degree of substitution (DS), which is the average value of the hydroxyl group in the

cellulosic units substituted by acetyl groups, is one of the most important parameters to evaluate CA [9]. In addition, DS significantly affects the polymer crystallinity and solubility in different solvents [10]. The solubility of CA in solvents such as acetone, chloroform, dichloromethane, and DMSO depends on the DS values [11], its value can vary from 0 (cellulose) to 3 (cellulose triacetate, CTA). In the case of the formation of cellulose diacetate (CDA), DS values may vary between 2.0–2.5 [12]. DMSO can dissolve diacetate and triacetate cellulose at room temperature.

Asymmetric membranes consist of a relatively dense thin layer, responsible for the separation, and a porous layer, often working as the mechanical support. To produce asymmetric membranes, the phase inversion method is often employed. When a substrate is immersed into a coagulation bath, there is an exchange between the solvent in the casting solution film and the non-solvent in the precipitation medium, leading to phase separation [13]. Ultrafiltration (UF) membranes have been widely employed in order to eliminate colloidal materials, suspended solids (SS), oil droplets, bacteria, and macromolecules for water purification [14,15]. Membranes fabricated from CA have a moderate flux, a high salt rejection, and are non-toxic [4,16,17].

Several publications reported the synthesis of CA from SB applied to membrane preparation for desalination, gas separation or drug release [4,16–18]. Candido et al. reported that lignin and hemicellulose remaining in extracted cellulose do not negatively affect the properties of the CA and obtained membranes. Manjarrez Nevárez et al. also reported that the functional groups in the lignin not only increase the porosity but also enhance the hydrophilicity of the cellulose triacetate substructure [19]. Thus, lignin remaining in synthesized cellulose acetate can also improve the porosity and hydrophilicity because of its abundant hydrophilic groups. However, the solvents used for membrane preparation from sugarcane bagasse are mostly toxic to human health and to the environment; typical examples are chloroform and dichloromethane. It is highly desirable to replace traditional toxic solvents such as chloroform, N-dimethylformamide (DMF), dichloromethane and dimethylacetamide (DMAc) by greener solvents for the preparation of polymeric membranes. DMSO, which can be extracted from lignin or synthesized by oxidation of dimethyl sulfide, is an aprotic organic solvent for membrane synthesis [20–22]. DMSO not only has a relatively low intrinsic toxicity, but also is biodegradable, forming non-toxic products [23]. Therefore, DMSO is considered a greener solvent for the fabrication of CA membranes.

In this work, a more suitable solvent for fabricating asymmetric membranes is introduced using sugarcane bagasse as a starting material, aiming at a good solubility and environmental friendliness. The prepared membranes have been investigated by studying their performance in ultrafiltration.

## 2. Materials and Methods

### 2.1. Materials and Chemicals

Sugarcane bagasse (SB) was collected from the sugar mill Binh Dinh (BISUCO), Viet Nam. Bovine serum albumin (BSA, 67 kDa), sodium hydroxide (NaOH, 98%), acetic acid (CH$_3$COOH, 99%), dimethyl sulfoxide (DMSO, ≥99%), sulfuric acid (H$_2$SO$_4$, 95–98%), acetic anhydride ((CH$_3$CO)$_2$O, 99%), EDTA, H$_2$O$_2$ (≥30%) have been purchased from Sigma-Aldrich (Diegem, Belgium).

### 2.2. Extraction of Cellulose from Sugarcane Bagasse

SB was dried in sunlight for four days until the humidity reached 5–8%; then it was cut into 1–2 cm pieces, which were milled and screened to pass through a 0.5 mm size screen. The milled SB was treated with chemical methods to obtain extracted cellulose. In the first step, dry SB powder (40 g) was boiled in distilled water for 2 h to dewax and remove solubilized hemicelluloses. The solid fragments were separated by vacuum filtration through Buchner funnel until a colorless filtrate was obtained. In the second step, the dewaxed SB was treated with sodium hydroxide (5%, *w/v*) with a ratio of liquid/solid 15:1 (mL/g) and heated to 80 °C for 2 h under vigorous stirring. Subsequently, the slurry was cooled to 50 °C, filtered and washed with distilled water by filtration using a vacuum pump until

the filtrate changed from brown-black to colorless. In the third step, the separated solid was chelated with 0.5% EDTA for 30 min at 70 °C with a solid to liquid ratio of 1:15 (g/mL). The residues were then filtered and washed with distilled water at 70 °C until the excess EDTA was completely removed. In the fourth step, chemical bleaching was conducted at two different agent concentrations and processing times. $H_2O_2$ was chosen as a bleaching agent for both ways. The chelated SB was bleached using 5% and 2% $H_2O_2$ solutions with pH value 12 (adjusted with concentrated NaOH solution) and a ratio of liquid to material 15:1 and stirring for 4 h at 70 °C. The bleaching was carried out once with 5% $H_2O_2$ and twice with 2% $H_2O_2$. After bleaching, the materials were filtered and washed with distilled water at 70 °C until neutral pH and excess $H_2O_2$ was removed completely. Finally, the extracted cellulose was dried in an oven overnight at 105 °C. The cellulose extracted from bleaching with 5% $H_2O_2$ was denoted as CE-1 and the cellulose extracted from bleaching with 2% $H_2O_2$ was denoted as CE-2.

### 2.3. Synthesis of Cellulose Acetate

CA was successfully prepared by the reaction between extracted cellulose and acetic anhydride, in the presence of $H_2SO_4$ as a catalyst following Cerqueira et al. [2], with modifications. At first, 50 mL acetic acid was added to 2.1 g extracted cellulose (CE-1 or CE-2) and stirred at 35 °C. After 45 min, a solution composed of 0.16 mL $H_2SO_4$ and 18 mL acetic acid was mixed to the system, which was stirred for 60 min at 35 °C. Afterwards the mixture was cooled down to 14 °C before adding 56 mL of acetic anhydride and 0.14 mL of $H_2SO_4$. Then, the mixture was heated to 35 °C and stirred for 30 min before turning off the magnetic stirrer and stand for 14 h at room temperature. After this time, the solution was centrifuged at 6000 rpm for 30 min to remove the undissolved solid. Thereafter, the liquid obtained was added into 2 L distilled water under stirring to stop the reaction and precipitate CA. The prepared CA was then filtrated and washed with distilled water by vacuum filtration to remove excess acetic acid until neutral pH. Finally, the produced cellulose acetate was dried at room temperature overnight. CA-1 and CA-2 were labeled for CA synthesized from CE-1 and CE-2, respectively.

### 2.4. Determination of Degree of Substitution and the Viscosity-Average Molecular Weight

The degree of substitution (DS) of acetylated samples was determined by titration with an aqueous sodium hydroxide solution as described in [24] and by $^1$H-NMR spectroscopy.

The viscosity-average molecular weight was determined as described by Cerqueira et al. [2]. Accordingly, the CA samples were dissolved in the dichloromethane/ethanol (*v/v* 8:2) solvent system with a concentration of 2 g/L. Ostwald viscometer was used to count the flow time of solvent system and polymer solution. The experiments were repeated three times to obtain accurate results. The viscosity-average molecular weight $\left(\overline{M}_v\right)$ was calculated by Equation (1):

$$\overline{M}_v = \left(\frac{[\eta]}{k}\right)^{\frac{1}{\alpha}}$$

(1)

where $k$ and $\alpha$ are constants depending on the type of solvent, polymer and temperature, $k = 13.9 \times 10^{-3}$ mL·g$^{-1}$ and $\alpha = 0.834$ in case of the dichloromethane/ethanol (*v/v* 8:2) solvent system at 25 °C. $[\eta]$ is the intrinsic viscosity, which is determined by Equation (2):

$$[\eta] = \frac{\sqrt{2(\eta_s - ln(\eta_r))}}{C}$$

(2)

where $\eta_r$ presents the relative viscosity ($\eta_r = \frac{t}{t_o}$); $t_o$ and $t$ are the flow time of solvent system and polymer solution, respectively; $\eta_s$ presents the specific viscosity ($\eta_s = \eta_r - 1$); and $C$ presents the concentration of the polymer solution.

### 2.5. Preparation of Asymmetric Cellulose Acetate Membranes

Asymmetric CA membranes were fabricated by phase inversion with a polymer concentration of 18 wt% in DMSO and DMSO/acetone (*v/v* 4:1) as solvents; these were named CAD and CADA membranes. At first, CA-2 was added in solvents and continuously stirred at 50 °C for 8 h to achieve a

homogeneous solution. After that, these solutions were placed in an oven at 50 °C for 4 h to remove gas bubbles. Subsequently, the cast solution was spread on a flat glass plate with a film thickness of 250 μm and was immediately immersed in the distilled water bath for precipitation. After the primary phase inversion, in order to ensure complete phase inversion, the obtained membranes were transferred to a new distilled water bath for 24 h. Finally, the synthesized membranes were rinsed and kept in distilled water for testing and characterization.

## 2.6. Characterizations and Analysis Methods

The functional groups of the materials were analyzed using an ATR-FTIR spectrometer (Bruker, Karlsruhe, Germany). X-ray diffraction (XRD) measurements were carried out using a D8 Advanced (Bruker, Karlsruhe, Germany), anode X-ray Diffractometer with Cu K$\alpha$ ($\lambda$ = 1.5406 Å) radiation. The thermal analysis of cellulose acetate and asymmetric membranes was made via DSC and TGA using Labsys Evo at a heating rate of 10 °C/min in a temperature range from 30 to 500 °C under argon flow (20 mL/min). The surface and cross-section morphologies of as-prepared membranes were studied by SEM (Nova nano SEM 450 FEI, Eindhoven, Netherlands). Surface properties of the membranes were analyzed with a Dimension Fast Scan AFM device (Bruker, Karlsruhe, Germany), in which the measuring area of each membrane sample was 1 μm × 1 μm. A contact angle goniometer (OCA20, Data physics Instruments GmbH, Filderstadt, Baden-Württemberg, Germany) was used to test the water contact angles of the membranes. The as-prepared membranes were immersed in deionized water at room temperature for 24 h to determine the water content. Afterwards, the surface of the membranes was mopped with tissue paper and weighed immediately. Then, these membranes were dried in a vacuum oven at 60 °C for 12 h, and the dry membranes were weighed. After that, these dry membranes were again placed in a vacuum oven until constant weight. The water content ($W_C$) was calculated by Equation (3):

$$W_C = \frac{W_w - W_d}{W_w} \times 100\%$$ (3)

where $W_w$ and $W_d$ are the weights of wet and dry membranes, respectively.

## 2.7. Membrane Performance of the Permeate Flux and the Antifouling Experiment

The permeability and rejection experiments were carried out with CA membranes using cross-flow filtration. The membranes had an effective area of 22.9 cm$^2$. Membranes were pre-compacted with a 0.15 MPa for 30 min. After that, the transmembrane pressure (TMP) was lowered to the operating pressure of 0.1 MPa. The pure water flux (J, L·m$^{-2}$·h$^{-1}$) was calculated by Equation (4):

$$J = \frac{V}{A.t}$$ (4)

where $V$ (L) represents the volume of permeate water, $A$ (m$^2$) the effective membrane area, and $t$ (h) the permeation time.

The resistance against fouling and the flux recovery ratio (FRR) of the prepared membranes were evaluated as follows: After measuring the pure water flux ($J_{w1}$), 500 ppm BSA solution as the protein foulant model was replaced and the permeate flux of BSA solution was recorded ($J_P$). After permeation of the BSA solution for 60 min, the fouled membranes were rinsed by distilled water for 20 min on a shaking table (150 rpm, 25 °C). Then, the flux of the cleaned membranes ($J_{w2}$) was measured again for 30 min by using Equation (2). After measuring the pure water flux, the BSA solution flux test and pure water flux were measured once again. The FRR was evaluated as exhibited by Equation (5), to determine the antifouling characteristics of the membranes.

$$FRR(\%) = \left( \frac{J_{w2}}{J_{w1}} \right) \times 100$$ (5)

A UV-vis spectrophotometer (PerkinElmer, Waltham, Massachusetts, USA) was used to measure the BSA concentrations in the feed and the permeate solutions at a wavelength of 280 nm. The BSA rejections ($R$) were determined by Equation (6):

$$R(\%) = \left(1 - \frac{C_p}{C_f}\right) \times 100 \tag{6}$$

where $C_p$ and $C_f$ (mg/mL) are the BSA concentrations in the permeate and feed solution, respectively. The measurements of the permeation and rejection were repeated for three different membranes to obtain accurate results and minimize the experimental error. To describe the fouling resistance, the total fouling ratio ($R_t$), reversible fouling ratio ($R_r$), and irreversible fouling ratio ($R_{ir}$) were calculated using the following equations:

$$R_t(\%) = \left(1 - \frac{J_P}{J_{w1}}\right) \times 100, \tag{7}$$

$$R_r(\%) = \left(\frac{J_{w2} - J_P}{J_{w1}}\right) \times 100, \tag{8}$$

$$R_{ir}(\%) = \left(\frac{J_{w1} - J_{w2}}{J_{w1}}\right) \times 100 = R_t(\%) - R_r(\%). \tag{9}$$

## 3. Results and Discussion

### 3.1. Extraction of Cellulose from Sugarcane Bagasse

#### 3.1.1. Chemical Composition

The chemical composition of cellulose isolated compared to SB was determined according to Vieira et al. [25]. The chemical composition of SB before treatment was 48.58% of cellulose, 23.69% of hemicellulose, 24.12% of lignin, and 3.61% of ashes. The results of extracted cellulose are shown in Table 1. The chemical composition of the extracted cellulose shows that a material with high purity has been obtained, in which the cellulose fraction increased from 48.58% to 83.05% (in CE-1) and 89.06% (in CE-2) compared to the SB composition. In the bleaching stage, hemicellulose and lignin have been mainly solubilized. The removal of lignin has been more effective with $H_2O_2$ at pH = 12 in the presence of NaOH. From the obtained results, bleaching with a low concentration of $H_2O_2$ (2%) has demonstrated that the isolated conditions were appropriate and effective; the cellulose content of the extracted cellulose is 89.09%, and the yield of overall extraction process is 66%.

**Table 1.** Chemical composition of extracted cellulose.

| Samples [a] | H$_2$O$_2$ Treatment | | | Result | | | |
|---|---|---|---|---|---|---|---|
| | Concentration of H$_2$O$_2$ (%, v/v) | Time (h) | Weight [b] (g) | Cellulose wt (%) | Hemicellulose wt (%) | Lignin wt (%) | Yield (%) |
| CE-1 | 5 | 4 | 6.0 | 83.05 | 8.25 | 7.94 | 60% |
| CE-2 | 2 | 4 for once | 6.6 | 89.09 | 7.11 | 3.07 | 66% |

[a] The samples obtained after dewax, alkali, and EDTA treatment. [b] Weight (g) of the extracted cellulose. Initial weight of raw sugarcane bagasse = 10 g.

#### 3.1.2. Fourier Transform Infrared Spectroscopy (FT-IR)

The efficiency of cellulose isolation is also expressed by the FT-IR spectra of raw SB and of extracted cellulose. Figure 1 shows the FT-IR spectra of the initial SB and extracted cellulose. The significant solubilization of lignin is represented by the absence of the peak at 1512 cm$^{-1}$, which is associated with C–C stretching of the aromatic skeletal vibration in lignin [5,26]. Furthermore, the intensity decrease of the peak at 1246 cm$^{-1}$ is ascribed to the characteristic C–O stretching of guaiacyl rings in lignin [27,28].

The disappearance of peak at 1743 cm$^{-1}$ assigned either to the acetyl and uronic ester groups of the hemicelluloses or to the ester linkage of carboxylic group of the ferulic and p-coumeric acids of lignin and/or hemicelluloses [29,30]. Simultaneously, the increase in the bands at 3412 cm$^{-1}$ and 2902 cm$^{-1}$, respectively, is assigned to the presence of the stretching of O–H and C–H bonds, which are typical bands of cellulose molecules [7]. The peak at 897 cm$^{-1}$ is associated to the β (1–4) linkages between the glucose monomers [31]. The intensity of the mentioned peaks increases as a function of the purity of cellulose in the obtained materials.

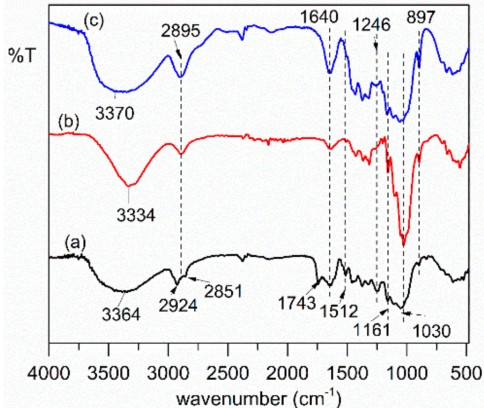

**Figure 1.** Fourier transform infrared spectroscopy (FT-IR) of (**a**) raw sugarcane bagasse (SB) and extracted cellulose from SB: (**b**) cellulose extracted from bleaching with 2% H$_2$O$_2$ (CE-2), (**c**) cellulose extracted from bleaching with 5% H$_2$O$_2$ (CE-1).

*3.2. Characterization of Cellulose Acetate*

3.2.1. FT-IR

FT-IR spectroscopy has also been used to examine the acetylated products. The FT-IR spectra of CA-1 and CA-2 are shown in Figure 2, in which five main changes in the cellulose acetate structure are visible. Firstly, a rise of the peaks at 1738 cm$^{-1}$ and 1224 cm$^{-1}$ assigned to the stretching of C=O ester carbonyl and –CO– bond in the acetyl group, respectively [17], has been observed. A peak at 1369 cm$^{-1}$ was observed, attributed to the bending vibration of the C–H bond in the acetyl groups [4]. The decrease of the band at 3400 cm$^{-1}$ relates to the replacement of the hydroxyl groups for the acetyl groups in the cellulose structure. The decrease in the intensity of the peak at 1634 cm$^{-1}$ indicates that CA represents a hydrophobic nature [7]. The peak at 1032 cm$^{-1}$ is characteristic of C–O–C pyranose ring skeletal vibration. The complete removal of acetic acid and acetic anhydride in the CA synthesized was confirmed by the absence of bands at 1760–1840 cm$^{-1}$ and 1700 cm$^{-1}$ [17].

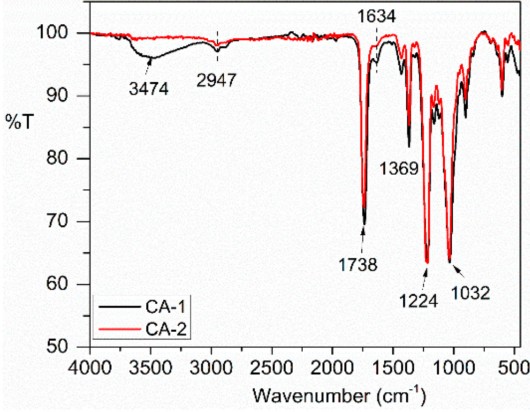

**Figure 2.** FT-IR of cellulose acetates (CA-1 and CA-2) synthesized from CE-1 and CE-2.

### 3.2.2. Degree of Substitution and Viscosity Average Molecular Weight

The acetylation of extracted cellulose resulted in acetylated products with DS value, average molecular weight ($M_w$) and ester yield (%) as summarized in Table 2.

**Table 2.** DS, % AG and average molecular weight of as-prepared cellulose acetate.

| Sample | % Acetyl Group (AG) | Degree of Substitution (DS) [b] | Viscosity η (mL/g) | Average Molecular Weight $M_v$ (g/mol) | Yield (% Dry Weight) [a] |
|--------|--------|--------|--------|--------|--------|
| CA-1 | 41.68 | 2.65 | 103.2 | 43,760.8 | 45 |
| CA-2 | 43.05 | 2.80 | 112.8 | 48,685.9 | 51 |

[a] Increase weight of the sample. [b] DS values from titration.

The data shown in Table 2 indicate that the rate of acetylation (%) and the DS value increased as a function of the purity of extracted cellulose.

### 3.2.3. [1]H-NMR Spectroscopy

[1]H-NMR spectroscopy was used to determine the distribution of acetyl among the three hydroxyl groups of the anhydroglucose, and to determine the DS value. The [1]H-NMR spectra for synthesized CA samples recorded in CDCl$_3$ are presented in Figure 3. The chemical shifts have been reported in parts per million (δ) from tetramethylsilane as an internal standard. Peaks between 1.9 and 2.2 ppm are assigned to three methyl protons of the acetyl groups. The presence of these peaks demonstrates that the extracted cellulose was successfully acetylated. The reaction order of three hydroxyl groups at $C_2$, $C_3$ and $C_6$ positions is $C_6$–OH > $C_2$–OH > $C_3$–OH, which is in good agreement with the previous reports [32]. In the [1]H-NMR spectrum of CA-1 (Figure 3a), the integrals for the peak areas are 2.82, 2.64 and 2.54 for $C_6$, $C_2$, and $C_3$, respectively. The total peak integral for all protons on the NMR spectrum for acetyl groups is 8.00. Peaks at 3.5–5.2 ppm relate to the seven protons of the anhydroglucose. The total peak integral for all protons on the NMR spectrum for the anhydroglucose is 6.99. Similarly, from the [1]H-NMR spectrum of CA-2 shown in Figure 3b, the total peak integral for all protons in the acetyl groups and for all protons in the anhydroglucose is 8.88 and 7.37, respectively. DS values are calculated from the [1]H-NMR spectroscopy results following Equation (10) [33]:

$$DS = \frac{7 \times I_{acetyl}}{3 \times I_{H,AGU}} \tag{10}$$

where $I_{acetyl}$ is the peak integral of methyl protons of acetyl groups, and $I_{H,AGU}$ is the peak integral of all protons of an anhydroglucose unit. The calculated DS values are 2.67 for CA-1 and 2.81 for CA-2, which are well consistent with the obtained values by the titration.

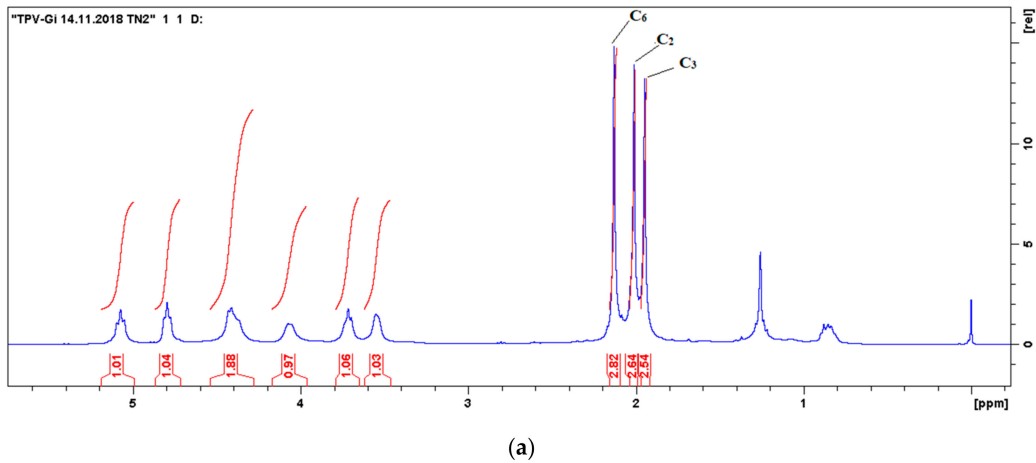

(a)

**Figure 3.** *Cont.*

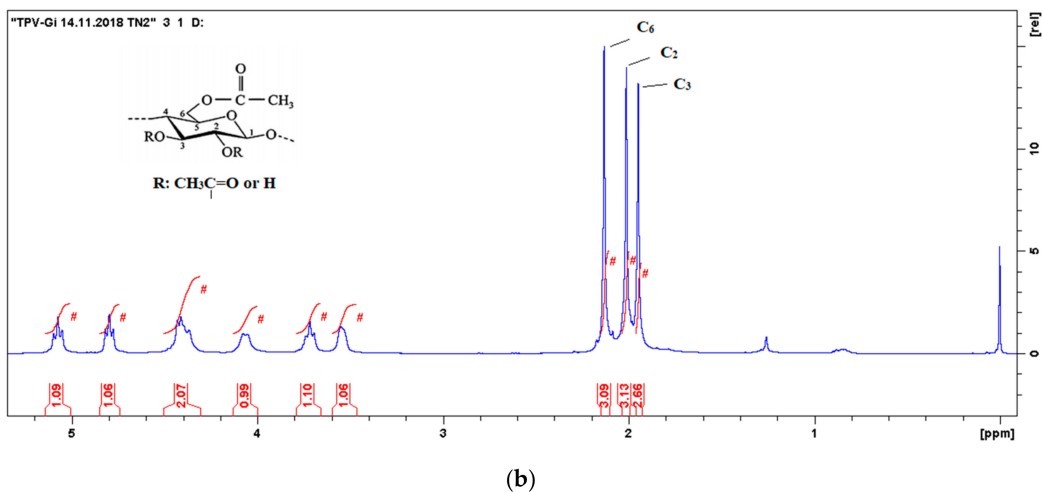

(**b**)

**Figure 3.** $^1$H-NMR spectra for (**a**) CA-1 (DS = 2.67) and (**b**) CA-2 (DS = 2.81).

### 3.2.4. XRD of Extracted Cellulose and Cellulose Acetate

X-ray powder diffraction was used for the determination of cellulose crystallinity. The crystallinity index (CrI) was calculated by the following equation [34]:

$$CrI = \frac{(I_{200} - I_{am})}{I_{200}} \times 100 \tag{11}$$

where $I_{200}$ is the intensity of the peak in the crystalline plan 200 (2θ = 22.2°) and $I_{am}$ is the intensity of the peak in the amorphous portion (2θ = 18.7°).

The XRD pattern of extracted cellulose and CA samples is shown in Figure 4. The XRD pattern of CE-1 and CE-2 (Figure 4a,b) show four diffraction peaks at 2θ = 14.8°, 16.0°, 22.2°, and 34.0° typical of cellulose crystal I (natural cellulose) [5,35]. These signals correspond to diffraction patterns 110, 101, 002 and 040. The shoulder diffraction peak at 16.0° and a weak peak at 34.0° indicates the absence of lignin and hemicellulose in the extracted cellulose [36]. CE-2 is purer than CE-1, which is demonstrated by the absence of a peak at 34.0°. This is also consistent with the obtained crystallinity index (CrI) values. CE-1 presents a CrI value of 55.27% and CE-2 presents a CrI value of 58.77%.

In Figure 4c, for CA-1 a strong peak appears at 8.0° and weak peaks at 10.3°, 12.4°, 16.6°, 21.5°, 22.2°, assigned to the crystalline peaks of cellulose triacetate II, while CA-2 (in Figure 4d) presents maxima at 7.6°, 17.6°, and 20.9°, corresponding to cellulose triacetate I [37]. In the acetylated process, cellulose acetate I was converted to cellulose acetate II, which may be because of the lignin content in CE-1.

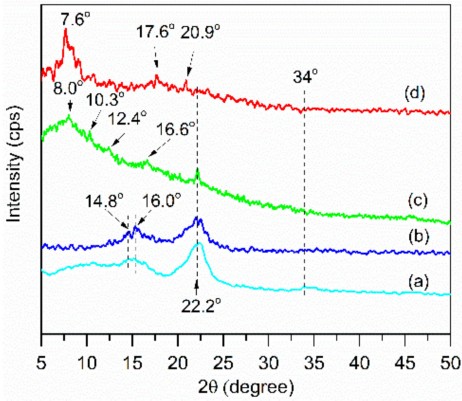

**Figure 4.** XRD patterns of extracted cellulose and CA samples synthesized from SB cellulose: (**a**) CE-1, (**b**) CE-2, (**c**) CA-1, (**d**) CA-2.

### 3.2.5. Differential Scanning Calorimetry-Thermogravimetric Analysis

The DSC and TGA curves of extracted cellulose and CA samples with different DS values are shown in Figure 5. The first endotherm of all samples occurs between 62.4 and 86.9 °C approximately in Figure 5a, corresponding to the desorption of water. However, each sample loses water at different temperatures, depending on the DS. The difference in the values of desorption peaks is attributed to the different water-holding capabilities and polymer–water interaction [8]. As can be seen from Figure 5a, the materials with lower DS have a higher desorption temperature. This is because the desorption process samples depend on the strength of interaction of water molecules with hydroxyl groups in the materials through hydrogen bonds. In the CA structure, OH groups are replaced by acetate groups, therefore rendering them less effective in their water holding capacity compared to pure cellulose [8,38]. The second endotherm relates to the melting of the CA samples; the melting temperatures are very close to one another, between 274.2 and 288.9 °C. The exothermic peaks correspond to the heat release from the compounds at a maximum temperature of 361.1 °C for CA-2 and 380 °C for CE-1, CE-2, and CA-1 because of the disintegration of intramolecular interaction and the decomposition of the polymer chain [5,39]. The highest rate of weight loss of CA occurs around 362.31 °C, which is higher than for extracted cellulose (around 328 °C) in the TGA curves (in Figure 5b). This clearly indicates that the acetylated cellulose has a higher thermal stability than native cellulose.

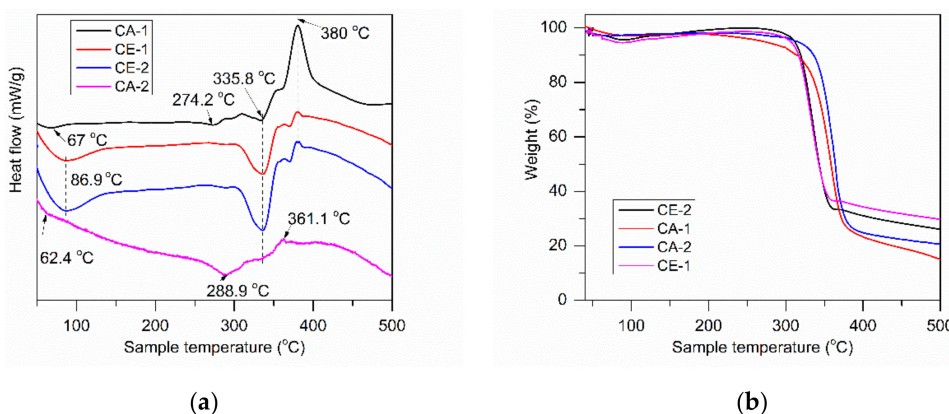

(**a**) (**b**)

**Figure 5.** Curves of (**a**) Differential Scanning Calorimetry (DSC) and (**b**) Thermogravimetric Analysis (TGA) to CE-1, CE-2, CA-1, and CA-2.

### 3.3. Characterization of Asymmetric Membranes

### 3.3.1. Scanning Electron Microscopy (SEM)

SEM images of the surface and cross-section of as-prepared membranes shown in Figure 6 indicate that all prepared membranes have an asymmetric structure with a dense thin top layer and a supporting layer with a finger-like and sponge-like porous structure. The high mutual diffusivity between water (non-solvent) and DMSO or DMSO/Acetone led the formation of an asymmetric structure during the phase inversion process [13]. The CADA membrane formed from DMSO/acetone solvent is more smooth and denser than CAD without acetone. Moreover, the interaction between lignin and hemicellulose remaining in CA with DMSO solvent resulted in a large number of pores on the top surface of the CAD membrane (Figure 6(a2)). In Figure 6(b2), fine cracks can be observed on the top surface of CADA membrane, which resemble spider webs. The formation of these cracks can happen when the top loses moisture so quickly during the drying process and may be due to residual stress between the dense top layer and the supporting layer of the ultrafiltration membrane. This surface cracking is also commonly found in ultrafiltration membrane with a double-layer structure [40–43]. The surface cracking was not observed in CAD membrane, which may be due to the surface of the CAD membrane having a lot of pores and more porosity. Cross-section images with higher magnifications of the bulk morphology are shown in Figure 6(c2,c3,d2,d3), in which the walls of macropores of the

CAD membrane are not smooth as those of the CADA membrane, but also more porous. Moreover, the micropores of CAD membrane (Figure 6(c2,c3)) are formed by the interlaced fibers as a saturated honeycomb structure.

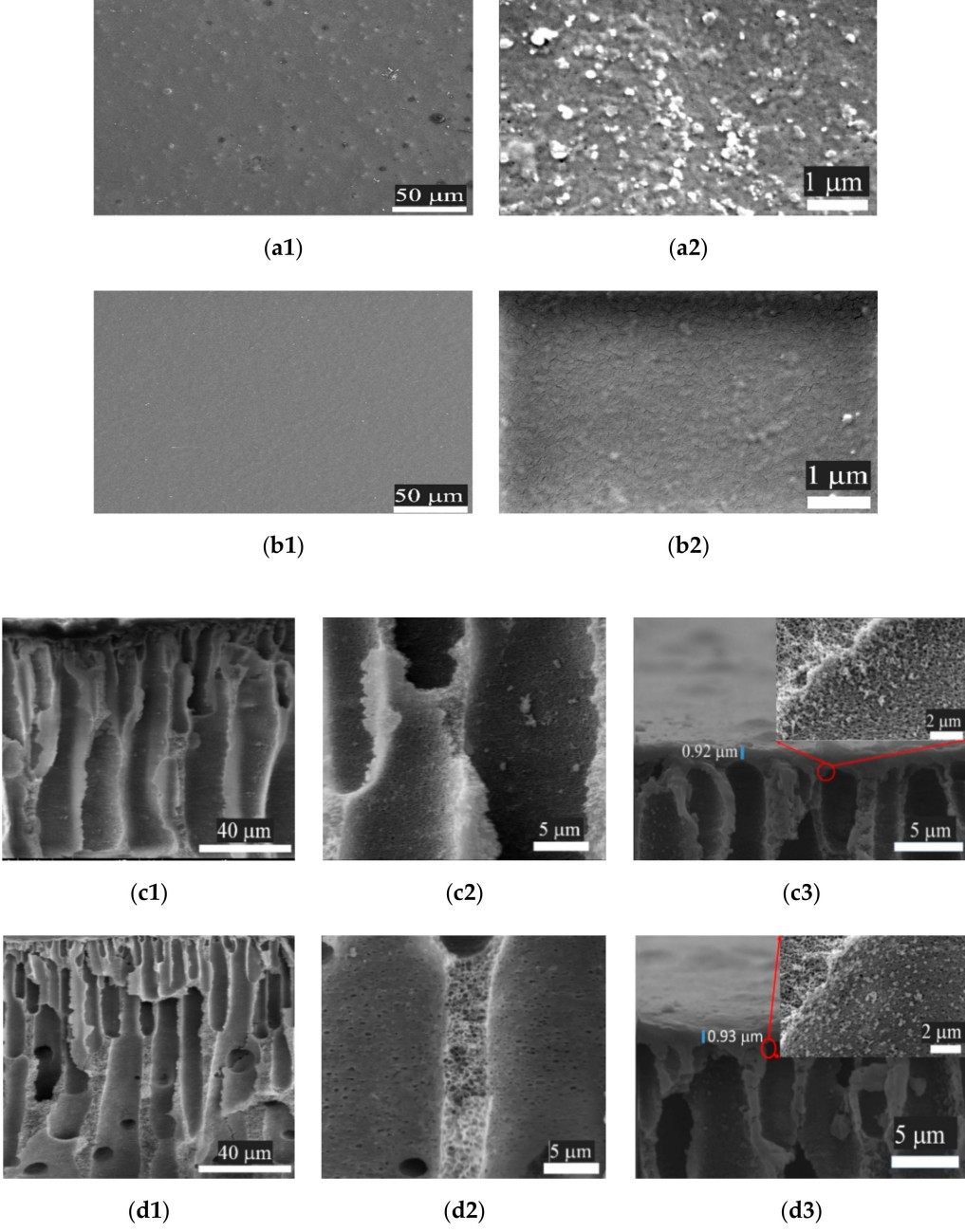

**Figure 6.** Scanning electron microscopy (SEM) images of the CA-DMSO (CAD) membrane (**a1,a2,c1–c3**) and the CA-DMSO/acetone (CADA) membrane (**b1,b2,d1–d3**).

### 3.3.2. DSC Analysis

DSC analysis has been used to study the thermal stability and the application capabilities of material in different temperature ranges, in which the glass transition temperature ($T_g$) is extremely important. The origin of glass transition is the motions of the segment around the backbone of the main chain in the case of linear polymers as the temperature increases. However, in hyperbranched polyesters, the movement of complete molecules governs $T_g$ [44]. As shown in Figure 7, the $T_g$ for the prepared membranes is in the range of 181–187 °C. Herein, the $T_g$ values are high due to the

formation intermolecular H-bond between polymers, which slightly prevents free motions of CA chains. The endothermic peaks at 60.1 and 55.4 °C are related to the desorption of water in materials. The endotherm of polymer fusion occurs at around 300 °C.

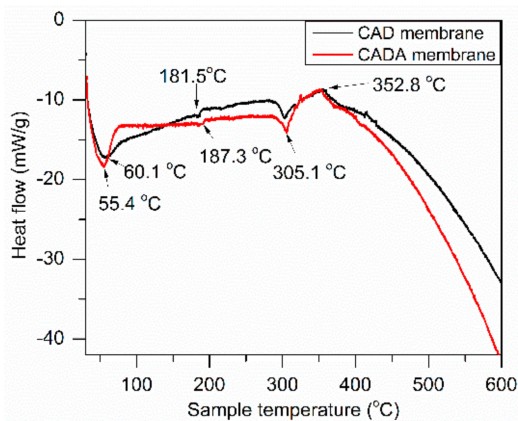

**Figure 7.** DSC curves of CAD and CADA membranes.

### 3.3.3. Atomic Force Microscopy (AFM) Analysis

AFM images of as-prepared membranes are shown in Figure 8. The dark area in the AFM images represents the valleys or membrane pores. In Figure 8a, dark regions are observed on the surface of the CAD membrane, which indicates that the membrane has pores in its structure. Moreover, some bright areas can be distinguished, which are attributed to the highest nodules or points [4]. From the data in Table 3, the CAD membrane has a rougher surface than CADA membrane, which is consistent with SEM observations. The membranes fabricated from self-synthesized CA and DMSO solvent has a rough and porous surface, while with pure commercial CA a smooth and dense surface has been obtained; the surface morphology changes to porous and rough only when using additives in the casting solution [45,46]. The presence of hemicellulose and lignin in its structure has a similar effect as the use of additives and contributed to the fabrication of rough and porous membranes.

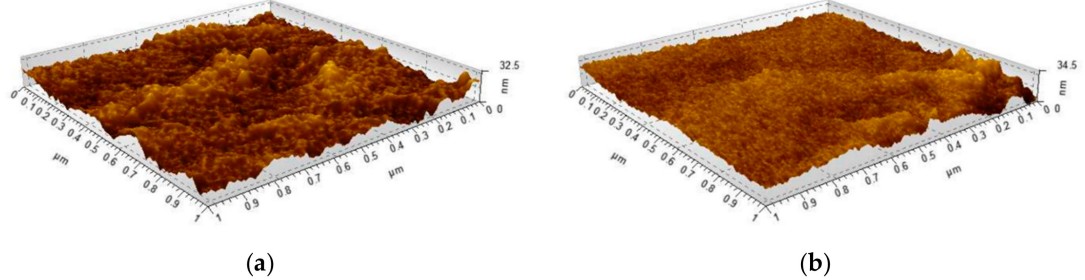

(**a**)                                                                    (**b**)

**Figure 8.** AFM micrographs of the CA membranes: (**a**) CDA membrane and (**b**) CADA (size of the micrographs: 1 µm × 1 µm).

**Table 3.** AFM Surface roughness parameters of the as-prepared membranes.

| Membrane | Root Mean Square Height (Sq) | Arithmetic Mean Height (Sa) |
|---|---|---|
| CAD | 4.45 nm | 3.51 nm |
| CADA | 3.34 nm | 2.53 nm |

### 3.3.4. Contact Angle and Content Water Measurements

The surface hydrophilicity plays an important role in liquid filtration [47]. Contact angle measurements are considered the most effective way for the evaluation of the relative hydrophilicity of a polymer membrane. From Figure 9 and Table 4, the contact angles of CAD and CADA are 55.6 ± 1.6°,

and 58.4 ± 1.8°, respectively, which indicate the hydrophilicity of the obtained membranes. The water content has been determined to depict the porosity, hydrophilicity and filtration performance of the membranes. The water content of membranes is higher than 75%, demonstrating their porosity, in which the CAD membrane has a slightly higher water content than the CADA membrane.

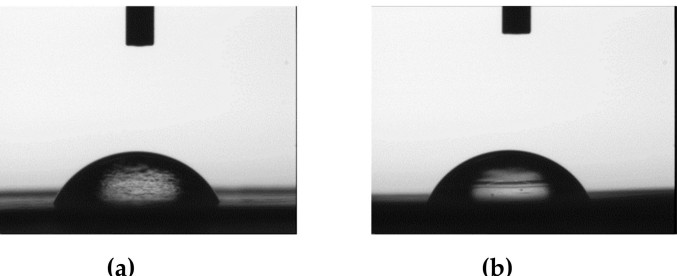

**(a)**                    **(b)**

**Figure 9.** A snapshot of the contact angle of prepared membranes: (**a**) CAD and (**b**) CADA membranes.

**Table 4.** Water content and contact angle of as-prepared membranes.

| Membranes | $W_C$ (%) | Contact Angle (°) |
|:---:|:---:|:---:|
| CAD | 78.34 | 55.6 ± 1.6 |
| CADA | 75.09 | 58.4 ± 1.8 |

### 3.3.5. Performance and Fouling Evaluation of the Membranes

The performance and rejection of the prepared membranes were studied through filtration experiments. The permeate flux of two membranes at a pressure of 0.1 MPa is summarized in Table 5. The results match with the results of the contact angle and water content tests, which indicate the hydrophilicity and the high flux of the prepared membranes. The BSA rejection of the CAD membrane is 85.7% and that of the CADA membrane is 87.3%. These results are better than the standard CA membranes which synthesized from commercial CA and NMP solvent in publication [48] as well as DMAc solvent [49].

**Table 5.** Summary of the pure water flux, permeation bovine serum albumin (BSA) flux, and water flux after cleaning of as-prepared membranes.

| Membrane | Pure Water Flux, $J_{w1}$ ($L\ m^{-2}\ h^{-1}$) | Permeation BSA Flux, $J_{p1}$ ($L\ m^{-2}\ h^{-1}$) | Pure Water Flux after Cleaning, $J_{w2}$ ($L\ m^{-2}\ h^{-1}$) | Permeation BSA Flux, $J_{p2}$ ($L\ m^{-2}\ h^{-1}$) | Pure Water Flux after Cleaning, $J_{w3}$ ($L\ m^{-2}\ h^{-1}$) |
|:---:|:---:|:---:|:---:|:---:|:---:|
| CAD | 314.41 | 136.24 | 288.21 | 128.38 | 248.91 |
| CADA | 301.31 | 131.01 | 248.91 | 99.56 | 196.51 |

A disadvantage in practical application of membrane technologies is the deterioration of permeability and selectivity because of membrane fouling. The antifouling properties of the synthesized membranes have been elucidated by a two-step filtration test (see experimental section for details) has been carried out using BSA as a model protein. A higher FRR and $R_r$ indicate that a large fraction of the fouling effect is reversible and the greater part of the flux could be recovered after the protein filtration process with hydraulic cleaning of weak foulant deposits [46,50]. The calculated flux recovery data are summarized in Table 6. This indicates that the prepared membranes have a great antifouling property, in which the CAD membrane has a much higher antifouling ability than the CADA membrane. Moreover, the FRR value of CAD and $R_r$ values are 80% and 35% higher, respectively, which indicates the high efficiency of hydraulic cleaning.

**Table 6.** Flux recovery ratio and resistance of CAD and CADA membranes.

| | First Cycle | | | | Second Cycle | | | |
|---|---|---|---|---|---|---|---|---|
| | FRR (%) | $R_t$ (%) | $R_r$ (%) | $R_{ir}$ (%) | FRR (%) | $R_t$ (%) | $R_r$ (%) | $R_{ir}$ (%) |
| CAD | 91.67 | 56.67 | 48.33 | 8.33 | 86.36 | 55.45 | 41.82 | 13.64 |
| CADA | 82.61 | 56.52 | 39.13 | 17.39 | 78.95 | 60.00 | 38.95 | 21.05 |

## 4. Conclusions

In conclusion, the results presented in this paper show that SB can be used as the starting material for synthesizing CA with the investigated synthesis conditions. CA showed a percentage of acetyl groups of 43.14% and a substitution degree of 2.81, i.e., cellulose triacetate was synthesized. The membranes which were prepared from self-synthesized cellulose acetate and DMSO show that it was possible to produce asymmetric membranes for application in separation techniques. The water contact angle and the performance results confirm the potential of these membranes obtained from sugarcane bagasse for this purpose. Moreover, the results also showed the suitability of DMSO solvent for membrane fabrication from naturally synthesized CA.

**Author Contributions:** Conceptualization, D.T.T.N.; Formal analysis, D.T.T.N.; Investigation, D.T.T.N.; Methodology, D.T.T.N.; Validation, N.P.H., C.V.H. and B.V.d.B.; Writing—original draft, D.T.T.N.; Writing—review & editing, N.P.H. and B.V.d.B.

**Funding:** This research was funded by VLIR-OUS (Belgium).

**Acknowledgments:** This work was financially supported by a VLIR-UOS (Belgium) TEAM project (code: ZEIN2016PR431).

**Conflicts of Interest:** The authors declare no conflict of interest.

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
