# Peer review of "Preparation of an Asymmetric Membrane from Sugarcane Bagasse Using DMSO as Green Solvent"

_applsci, doi:10.3390/app9163347_

Round 1

Reviewer 1 Report

Present paper deals with preparation of asymmetric membranes from natural sugarcane bagasse. Authors motivate their project by using natural “waste” to prepare and characterize separation membranes. Membrane preparation is done by carefully cleaning the natural products, extracting cellulose and transferring it to cellulose acetate. This raw material was then converted to membrane material by phase inversion. Membranes were characterized by XRD, FTIR, DSC and H-NMR spectroscopy. Also realistic permeation experiments including influence of fouling have been performed. The authors conclude that their approach results in suitable membranes for permeation.

The paper is reasonably written, figures, headline and language are o.k.

I have some comments

Fig. 7. DSC of the membranes. The shape of the DSC curves in the range around 100 °C does not resemble a typical glass transition. There a step is expected, not a peak. A peak around 100 C° is more likely related to water. Authors should repeat their measurements in that range on the same sample twice to check that.

Table 5: other data: it would be useful for the non-expert, to compare present data on permeation with similar systems or state of the art membranes.

Minor typo: table 1, last row, should be Yield, not Yeild

Summary: scientific approach is good, paper is well written,. Fits to journal should be accepted after minor, mandatory changes.

Author Response

Dear Reviewer,

Thank you so much for your kind comments.

It is very helpful to me. I have re-measured DSC of membranes to check and the results were presented in the manuscript.

I edited according to your comments. I hope that you can accept my manuscript after revision.

Thank you so much again.

Kind regards,

Dang Thi To Nu

Reviewer 2 Report

the work entitled as: "Preparation of an asymmetric membrane from sugarcane bagasse using DMSO as a green solvent"  is an interesting work dealing with the fabrication of new Cellulose acetate membranes, from natural extraction way.

It can be accepted after revision to the following points: Please provide better scalebar for the SEM images. Also is it possible to provide a close look to the top surface fo the crosssection of the membrane for a more detailed evaluation of the structure?

The authors are not presenting membrane separation performance. Cna the authors include also a part of this aspect?

Could there be better results obtained by the use of another solvent? the reason of green solvent use is understood, but in case that another solvent can provide 10 faster the results of DMSO that means that one is not necessary to use high solvent amount which regardless of green or not green classification it is organic water waste.

Author Response

Dear Reviewer, 

Thank you very much for your comments.

I  respectfully send the answers to you in attachment file.

I hope that you are satisfied with these answers.

Thank you so much again.

Kind regards,

Dang Thi To Nu

Reviewer 3 Report

I have read the manuscript “Preparation of an asymmetric membrane from sugarcane bagasse using DMSO as green solventby Dang Thi To Nu et al. (MS # applsci-559168) submitted for the publication in Applied Sciences.

The authors report their investigations on the preparation and characterization by several techniques (FT-IR, NMR, XRD, DSC-TGA, SEM, AFM, and contact angle measurements) of asymmetric membranes from sugarcane bagasse using DMSO as green solvent. The performance in ultrafiltration and antifouling properties of such membranes were also investigated.

Results are interesting but, in my opinion, the manuscript needs minor revisions before its publication in Applied Sciences. In particular:

1.     All acronyms should be defined the first time they are used in the text and not only in the abstract (e.g. page1 line 38 “cellulose acetate”).

2.     Page 2 line 70: the use of DMSO as green solvent in the preparation of PVDF membranes was recently reported in ACS Applied Polymer Materials 2019, 1, 3, 326-334. This paper should be added in the reference.

3.     Page 3 line 121: the determination of viscosity –average molecular weight should be briefly reported.

4.     Page 4 equation 7: Rt - Rr should be Rt(%) – Rr(%).

5.     Figure captions and Tables should be separated from the main text by an empty line.

6.     Please, specify that DS in the caption of Table 2 was obtained from titration.

7.     What about the well defined peak at 31.8° in Figure 4c?

8.     Size bars are not visible in all pictures of Figure 6.

9.     Please, round errors in contact angle measurements both in the main text and Table 4.

10.  Some misprints: “et al.” rather than “et al”, Yeild in Table 1, CA-1 and CA-2 rather than CA1 and CA2, diffraction peaks described in line 264 are four rather than three, Sample temperature rather than Temperature sample in the x-axis labels of Figures 5.

Author Response

Dear Reviewer,

Thank you so much for your kind comments.

We have added the paper reported in ACS Applied Polymer Materials 2019, 1, 3, 326-334 as you suggested in the reference. All your comments were edited in the manuscript.

The peak at 31.8o may be contamination during the XRD measurement. I will repeat the measurement to check.

Thank you so much again.

Kind regards,

Dang Thi To Nu.

Round 2

Reviewer 2 Report

The revised version of the manuscript does not meet yet the criteria for acceptance. Regarding the comments that I enclosed in the first version, there are only modifications for the scale bar of SEM images. There is no extra addition to the presentation of the results of the surfaces of the membranes, e.g., the explanation for the cracks that are identified there (figure 6b). Are these cracks artifact's or they can be attributed to some other phenomena? Additionally, cannot verify the honeycomb structure that the authors claim. Please include in the manuscript.

Regarding the membrane performance, the rejection experiments are explaining the performance of the separation but the comparison to another membrane used for this separation is missing.  This is a comment that the authors could include in the manuscript and will help another scientist to follow the authors' way for the membrane fabrication via naturally extracted CA.

Finally, the authors should add also a comment in the manuscript that only with DMSO they got the best results for the extraction of CA.

Author Response

Dear Reviewer,

Thank you so much for your kind comments. They are very helpful to me.

I have revised in the manuscript. 

I hope you accept the revision.

Kind regards,

Dang Thi To Nu.